# Coarse land cover datasets bias Arctic-Boreal wetland methane budgets
Josh Hashemi [1] ✉, Aleksi Räsänen [2], Tarmo Virtanen[3], Sari Juutinen [4], Guido Grosse [1,5], Mika Aurela [4], Annett Bartsch [6], Laura Chasmer [7], Scott J. Davidson[8,9], Mika Korkiakoski [4], McKenzie A. Kuhn [10], Mark J. Lara [11], Miska Luoto [12], Pekka Niittynen[13], David Olefeldt [14], Oliver Sonnentag [15], Anna-Maria Virkkala [16], Carolina Voigt[1,15,17] & Claire C. Treat [1,18]

Accurate methane ($CH_4$) emission estimates from Arctic and boreal wetlands are essential for reducing global budget uncertainties but are hindered by poorly constrained wetland distribution and classification. We assessed how land cover map resolution and thematic detail influence these estimates. Using very high spatial resolution land cover maps (≤2.5 m) with five to seven harmonized classes and 4–50% wetland coverage, we estimated $CH_4$ emissions across seven Arctic and boreal sites in North America and Eurasia. Resampling to coarser resolutions (up to 5 km) revealed that $CH_4$ flux estimates remained within 13% error when resolution was ≤25 m pixel size. At resolutions coarser than 1 km, four of seven sites shifted from net $CH_4$ source to sink, due to misrepresentation of wetland extent in heterogeneous landscapes with small, fragmented wetlands. Thematic detail also proved critical, as fens—high $CH_4$ emitters—were disproportionately underrepresented in coarse (>1 km) maps relative to other wetland types. We also show that existing global or circumpolar land cover maps tend to misrepresent wetlands, either overlooking smaller features or overestimating coverage in wetland dominated areas. Our findings indicate that coarse-scale land cover datasets are unsuitable for estimating $CH_4$ budgets in these regions, where high spatial resolution and biogeochemically relevant land cover classes are essential for reliable $CH_4$ emission upscaling.

Arctic and boreal wetlands, including peatlands, lowland tundra, and other waterlogged ecosystems, are significant sources of methane ($CH_4$), yet their contributions to the global budget remain poorly constrained, with estimates ranging from 8 to 55 Tg $CH_4$ $yr^{-1}$[1–4]. Considerable disagreement exists between bottom-up approaches[5] (field measurements and process-based modeling) and top-down approaches (atmospheric inverse modeling), particularly regarding natural $CH_4$ sources and emissions from permafrost regions[2]. Bottom-up estimates of $CH_4$ emissions at regional or global scales require both flux observations from field measurements, and spatial representation of $CH_4$ sources and sinks, often derived from land cover datasets[1,6–8]. Each of these components contribute to the overall uncertainty of $CH_4$ emission estimates[9] in a region with relatively sparse flux measurement density[4,10,11]. Recent studies have emphasized the importance of thematic detail (wetland type) for differences in $CH_4$ flux ($FCH_4$) rates, with larger emissions from minerotrophic wetlands, such as fens, and smaller emissions from ombrotrophic wetlands, such as bogs[3,11–13]. Fens are groundwater-fed, nutrient-rich wetlands, while bogs are precipitation-fed, nutrient-poor systems. Boreal and Arctic peatlands are often mosaics of these wetland types, where low-lying areas tend to be fens and elevated microforms are more often bogs. Despite the established link between regional emissions and wetland extent and type[14,15], the sensitivity of regional to global $CH_4$ budgets to uncertainty in the distribution of $CH_4$-emitting areas remains poorly understood[1,16].

Land cover classifications used to estimate areas of $CH_4$ emission and uptake[13,17–24] may vary widely in spatial and temporal resolution, thematic detail, and methodological approaches, contributing to uncertainty in $CH_4$ budgets[9]. Over recent decades, spatial resolution—the size of the pixels or grid cells that represent different land cover types—has significantly improved for wetland and lake extent maps[25]. However, resolutions are often still too coarse to capture the high heterogeneity of Arctic and boreal wetlands[14,15]. Fine-scale landscape features—such as ecotones, narrow channels, patterned ground, and isolated vegetation patches—require spatial resolutions commensurate with their characteristic dimensions (often <10 m) in order to be accurately represented, but these are generally obscured or simplified in coarser datasets[15,26–28]. For example, ecotonal transition zones at lake margins and shore vegetation often serve as $CH_4$ emission hotspots[29]. Similarly, areas of $CH_4$ uptake—underrepresented in current observations—require high-resolution mapping of wetland–upland

---

boundaries to improve estimates of $CH_4$ sinks in heterogeneous wetlands[30]. Another key consideration is that the wetland classification scheme used may not differentiate $CH_4$ emissions, often grouping areas with distinct flux dynamics. Global vegetation maps often misclassify Arctic landscapes, with broad categories such as "grassland" failing to distinguish between wet and dry tundra, which differ in $CH_4$ emissions[31]. Differentiating these wet and dry tundra landscape types can be challenging due to their similar structural and spectral characteristics[32,33] but is necessary to improve high-latitude $CH_4$ budgets[34].

Resolving fine-scale landscape details, essential for $FCH_4$, requires high spatial resolution data. While datasets at various spatial resolutions (20−1000 m) from spaceborne sensors like MODIS, Landsat, or Sentinel-2 are freely available on a global scale, very high spatial resolution data (≤2.5 m) from airborne sensors is often restricted to smaller regions. Commercially available products from providers like Planet Labs or Maxar offer high-resolution imagery but acquiring and processing these datasets for extensive areas, such as the entire circumpolar region, is time-consuming, resource-intensive, and currently cost-prohibitive. These challenges result in (1) studies focused on smaller regions that may not fully capture landscape variability[14,35–37], or (2) large-scale analyses using coarser-resolution imagery, which fails to represent fine-scale features critical for $FCH_4$[1,7,16]. These limitations underlie the need to determine a minimum spatial resolution required to accurately capture landscape variability and $FCH_4$ dynamics without compromising spatial coverage.

While spatial resolution and thematic detail are important considerations for pan-Arctic and boreal $CH_4$ budgets, there is a trade-off between accuracy and practicality for map resolutions in land cover products used in $FCH_4$ upscaling. Here, we synthesize very high nominal resolution (<2.5 m pixel size) land cover classification maps in conjunction with upscaled chamber-based measurements of $FCH_4$ at seven study sites from the Arctic and boreal region. Study sites included: Tiksi, Russia; Seida, Russia; Kilpisjärvi, Finland; Pallas, Finland; Utqiaġvik, United States; and two sites in Scotty Creek, Canada (North and South; Fig. 1). We assess how regional $FCH_4$ estimates are affected by spatial resolution by resampling land cover maps from fine to coarse scales (Fig. 1; Supplementary Fig. 1) and quantify how resolution-dependent errors relate to wetland size, type, and landscape fragmentation.

## Results and discussion
### Wetland distribution and fluxes with coarse resolution mapping
The study sites are distributed across the pan-Arctic and boreal, including North America, Russia, and Fennoscandia and range from boreal and sub-Arctic forest to Arctic tundra (Fig. 1). Wetlands were present across all the study regions; the median wetland fraction for all regions was ~35% (4–50%; Supplementary Table 1) at nominal resolution (Fig. 2a). Land cover was grouped to 6-7 classes at each site. Wetland classes were harmonized to represent total wetlands at all sites. At five sites (Pallas, Tiksi, Seida, and two sites at Scotty Creek), fen and bog classes were identified. Across these sites, bogs comprised approximately 18% and fens around 14% of mapped extents. For most regions (six of seven sites), median wetland extent decreased during resampling to coarser resolutions from 35% at the 2.5 m reference resolution to 32% coverage at 25 m resolution and 20% at 100 m resolution (Fig. 2a). By 250 m spatial resolution, the median wetland fraction had fallen by nearly half, to 22% coverage. At 1250 m resolution, the median wetland fraction across the landscapes reached zero (Fig. 2a).

A strong positive correlation ($R^2 = 0.6$) was found between regional upscaled estimates of $CH_4$ emissions and wetland area across the study sites (Fig. 2b). However, the unexplained variability in $FCH_4$ suggests that other factors—related to the thematic detail of land cover classification, such as variations in climate, vegetation community, permafrost presence, and nutrient regime—also contribute significantly to the observed emissions. Among the study sites, the region with the smallest wetland area (4.7%), Kilpisjärvi in sub-Arctic Finland, exhibited the lowest $FCH_4$ ($0.16 \pm 0.32$ g C-$CH_4$ ha$^{-1}$ h$^{-1}$), while South Scotty Creek in the boreal forest of the Taiga Plains ecozone of the Northwest Territories, Canada, with 50% wetlands, exhibited the highest $FCH_4$ ($21 \pm 4.8$ g C-$CH_4$ ha$^{-1}$ h$^{-1}$; Fig. 2b, c). The difference in estimated wetland extent at coarser resolutions had strong effects on the net regional $FCH_4$ emissions (Fig. 2c). In four of seven study regions, using coarser resolution classifications led to a switch from net $CH_4$ emissions to net uptake of atmospheric $CH_4$, particularly at or above 1 km grid size, while in other sites regional emissions decreased by 75.5–100% (Fig. 2c). The notable exception to decreasing $FCH_4$ using coarser resolution imagery was South Scotty Creek (Fig. 2c). Emissions from this region increased by 142% at the maximum pixel size, from $21 \pm 4.8$ g C-$CH_4$ ha$^{-1}$ h$^{-1}$ to $50 \pm 13.8$ g C-$CH_4$ ha$^{-1}$ h$^{-1}$, indicating that though wetland extent is driving $FCH_4$ (Fig. 2b), the non-contiguous spatial patterns of small wetlands are lost at scale.

Given the varying directional shifts in regional $FCH_4$ emissions among sites with decreasing spatial resolution (Fig. 2c), we calculated the absolute values of regional $FCH_4$ and expressed them as percent deviations from the nominal resolution $FCH_4$ (see Methods) to quantify the magnitude of scaling error introduced by coarser resolutions. Across all sites, deviations remained small at fine resolutions, with changes of only 6 % observed up to a 10 m pixel size (Fig. 3). However, beyond this point, error propagated

**Fig. 1 | Study Sites and Methodological approach.** (Left) site locations used in this synthesis (1 – Scotty Creek, Canada; 2 – Utqiaġvik, United States; 3 – Tiksi, Russia; 4 – Seida, Russia; 5 – Kilpisjärvi, Finland; 6 – Pallas, Finland). Coloration within the land region shows WAD2M wetland distribution for August, 2018. The blue line shows the boundary for the Arctic region as defined by the Conservation of Arctic Flora and Fauna (CAFF) working group. The yellow line shows the treeline. Dotted line shows the Arctic Circle (66° 34' N). Gray regions denote areas with no wetlands. (Right) example resolution coarsening of LC map of Tiksi, RU.

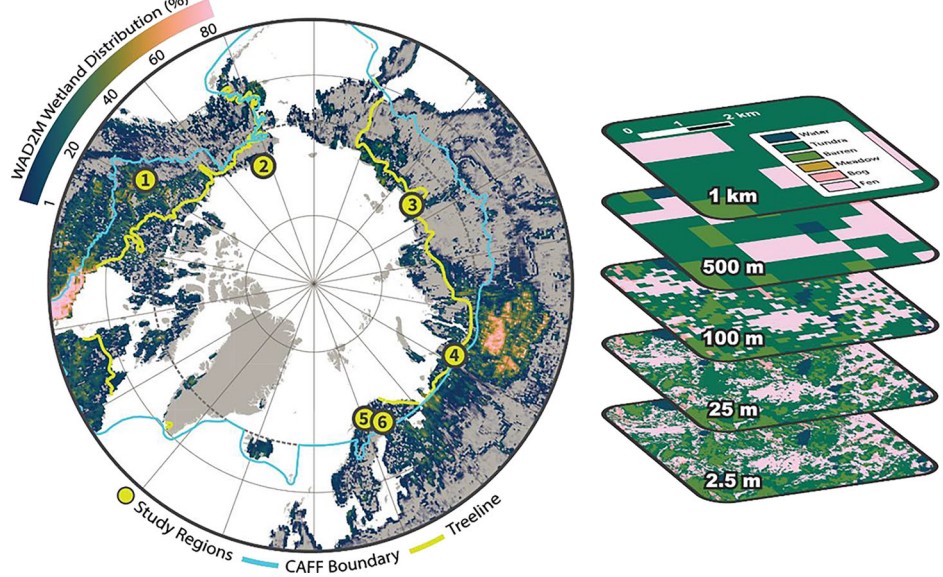

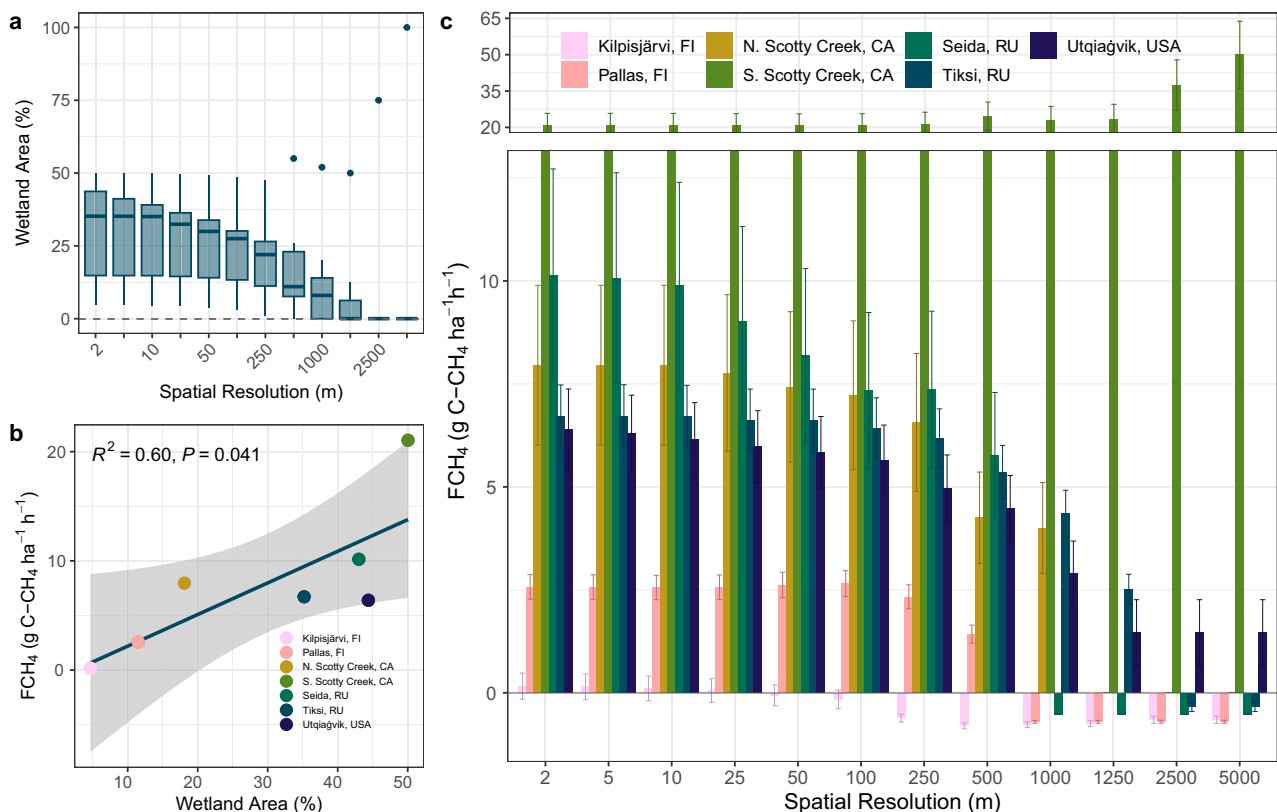

**Fig. 2 | Interaction of spatial resolution, wetland extent and FCH4 estimates across regions. a** Regional wetland area (% of mapped area) as a function of spatial resolution; **b** Linear regression of upscaled regional FCH4 (expressed as g C ha-1 h-1) and regional wetland area (%) at the nominal resolution (*n* = 7). Shaded area represents confidence interval for the fitted trend; **c** Magnitudes of total regional FCH4 (expressed as g C ha$^{-1}$ h$^{-1}$) by site at increasingly coarse resolutions. Error bars represent standard error of flux measurements.

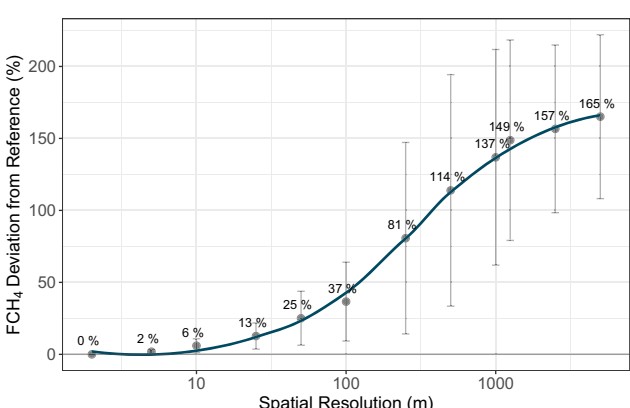

**Fig. 3 | Percent deviation in FCH$_4$ with decreasing spatial resolution.** Points and error bars represent means across sites ± standard error. Percent deviation is defined as the absolute change in FCH$_4$ expressed as a percentage of the FCH$_4$ at the nominal reference resolution. Higher values indicate larger error.

rapidly: at 50 m resolution, FCH$_4$ estimates had already diverged by 25 ± 19%. Coarser resolutions similar to many pan-Arctic and global land cover datasets—250 m and 1000 m—introduced substantial deviations of 81 ± 67% and 137 ± 75%, respectively (Fig. 3), well exceeding the average uncertainty (standard error) of the flux measurements at the highest resolution (43.7%; Fig. 2c). These results emphasize the critical need for geospatial data at spatial resolutions commensurate with emission source heterogeneity to minimize upscaling errors.

FCH$_4$ percent deviations were strongly linked to underlying landscape structure (Fig. 4). We characterized landscape structure using: (1) wetland patch size, quantified by mean patch size and the standard deviation of patch area (higher values indicate larger and more size-variable patches); and (2) fragmentation, represented by the landscape division index (higher values reflect more fragmented landscapes with numerous small, isolated patches). These metrics governed the sensitivity of FCH$_4$ estimates to decreasing spatial resolution. For example, the Kilpisjärvi region exhibited a CH$_4$ source-to-sink transition at a finer resolution (50 m) relative to the other sites (Fig. 2c), and the most rapid percent deviation from reference resolution FCH$_4$ (Supplementary Fig. 2). Kilpisjärvi had the highest landscape division index (Fig. 4a), smallest mean wetland patch areas (Fig. 4b), and lowest standard deviation of wetland patch area (Fig. 4c), consistent with the high micro- and meso-topographic variation at the site[36]. As resolution coarsened, smaller high CH$_4$-emitting patches were increasingly eliminated during pixel aggregation, leading to an apparent net CH$_4$ sink (Fig. 2c).

Conversely, regions like the South Scotty Creek site, with more aggregated wetlands (Fig. 4a) and larger wetland patch size (Fig. 4b), were less sensitive to patch loss at coarser resolutions, but instead exhibited inflated CH$_4$ estimates as wetland areas were overrepresented (Fig. 2c). Regions with smaller, more fragmented patches are generally more sensitive to FCH$_4$ errors introduced by coarse-resolution geospatial products, which is particularly important because these wetland types often contribute disproportionately to CH$_4$ emissions. While fragmentation can be influenced by classification method (e.g., pixel-based vs. object-based approaches and post-classification processing), it primarily reflects the underlying wetland types present. These classification effects are most relevant at the native resolution, where changes in CH$_4$ emission magnitudes (Fig. 2c) and FCH$_4$ percent deviation (Fig. 3) were minimal.

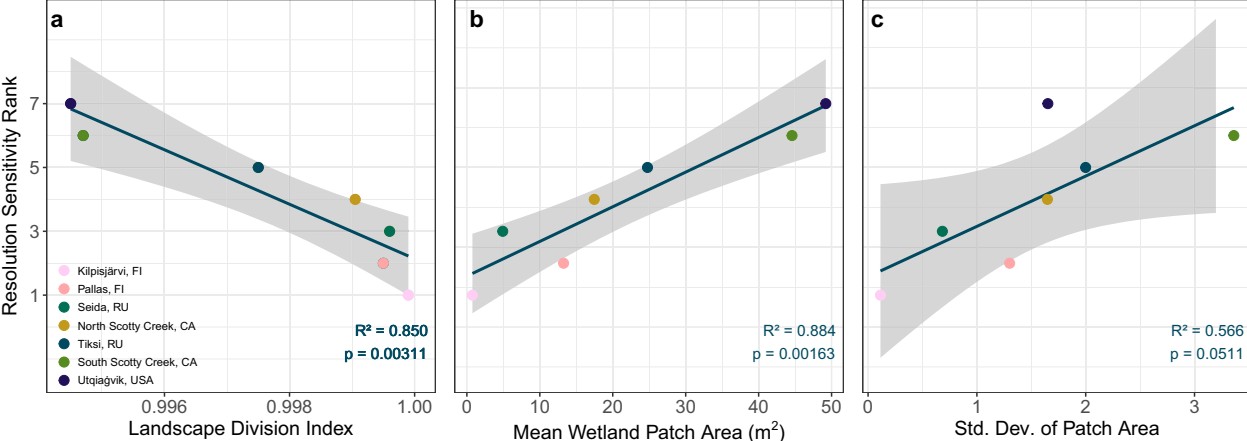

**Fig. 4 | Geomorphological drivers of resolution sensitivity.** Linear regressions ($n = 7$) of resolution sensitivity rank against (**a**) mean landscape division index, mean wetland patch are (**b**), and (**c**) standard deviation of patch area within each region. Resolution sensitivity rank indicates the relative sensitivity of a region's FCH4 signal to spatial resolution coarsening, ranked from 1 (most sensitive) to 7 (least). Ranks were derived as the area under the deviation-from-reference curve for each region (Supplementary Fig. 2). Areas were calculated using the trapezoidal rule over log-transformed resolution steps to normalize distance between resolutions. Shaded regions represent 95% confidence intervals.

**Fig. 5 | Mean proportional contributions of bog and fen to total FCH4 emissions across spatial resolutions.** Bars are stacked by wetland type. Error bars indicate ± standard error of each type's proportional contribution within each resolution.

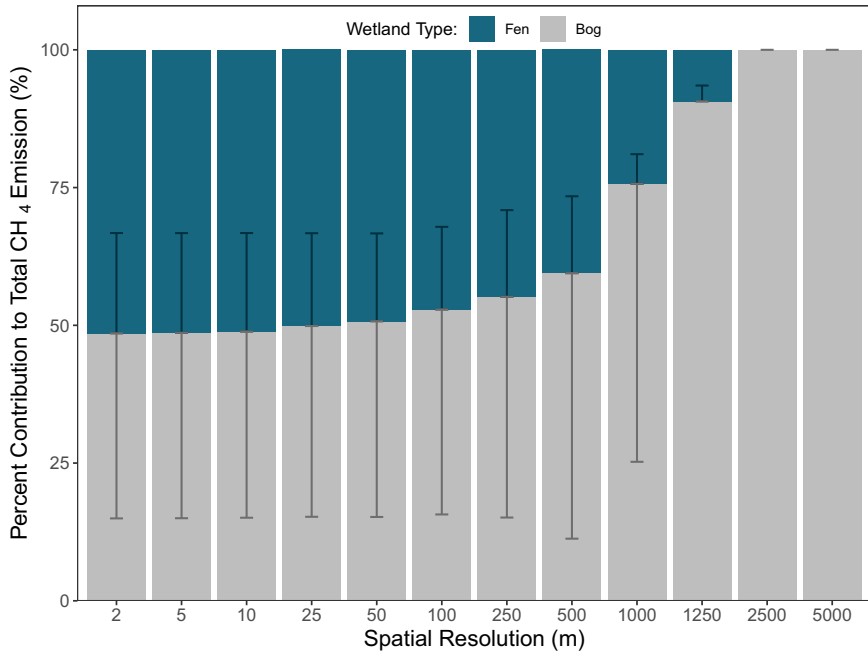

## Representation of wetland type and other methane-emitting areas

Thematic detail in wetland categorization is important because of the often markedly different $CH_4$ emission rates among wetland types such as bogs, fens, and permafrost wetlands[3,11]. Here, fens and bogs contributed equally to total $CH_4$ emissions at the reference resolution, but bog emissions outweighed fen emissions of $CH_4$ at >500 m spatial resolution (Fig. 5), with the contributions at the coarsest resolutions (>2.5 km) driven by the site that retained substantial wetland area (i.e., South Scotty Creek; Fig. 2c). Analysis of landscape metrics revealed clear structural differences between bogs and fens that drive this. The landscape division index was high for both wetland types (bogs: 0.9976; fens: 0.9986), indicating extensive fragmentation overall, though fens exhibited slightly higher division, suggesting a greater degree of patch isolation. Additionally, bogs exhibited larger mean patch sizes (24.13 m² for bogs; 17.87 m² for fens) and greater variability in patch size (standard deviation: 2.01 for bogs; 1.59 for fens), indicating a wider range of patch sizes within bog systems. Collectively, these metrics suggest that bogs form larger and more spatially extensive patches, whereas fens tend to consist of smaller, more uniformly sized, and highly fragmented patches embedded within a more heterogeneous landscape matrix. These morphological differences likely explain why bog $CH_4$ emissions were less affected by decreasing spatial resolution, as their larger, more extensive patch structures were better preserved during coarsening.

Despite their importance to FCH4, current land cover classifications often underrepresent small, morphologically complex wetland features and aquatic systems, such as polygon troughs over degrading ice wedges, graminoid meadows, aquatic macrophyte beds, and ponds. Lakes, which constitute an average of 1.7% of the reference maps, represent an additional

$CH_4$ source in the upscaled $FCH_4$ estimates as these areas can be significant local $CH_4$ emitters[3,38-41]. Due to a lack of in-situ measurements, values for lake fluxes were taken from the BAWLD-$CH_4$ database[3], corresponding to median "medium-sized peatland lakes". Even though lakes generally lack the same level of fragmentation and intricate morphology as wetlands, small features like shallow water-land edges or areas of emergent aquatic vegetation are hotspots for $CH_4$ emissions and therefore critical to capture[29,42-44]. Streams are often too sinuous to be accurately mapped in large-extent land cover classifications but may also release substantial amounts of $CH_4$[45].

## Implications for $CH_4$ budget estimates

Wetland distribution across the Arctic and boreal landscapes may be more extensive and densely concentrated than suggested by existing coarse-resolution wetland maps (Fig. 2a, Supplementary Fig. 3a–d), with significant implications for regional $CH_4$ emission estimates (Fig. 2c). Many current wetland maps may omit wetland areas or types in regions characterized by high spatial heterogeneity and/or where wetlands are not the predominant land cover type (Fig. 2a). However, they also likely overestimate wetlands in wetland-rich areas when coarser-resolution, continental- to global-scale wetland products are used. In this study, wetland extent dramatically increased for wetland-rich South Scotty Creek at coarser resolutions, doubling regional $CH_4$ emissions at the coarsest resolution (Fig. 2c). Comparisons between the reference wetland maps used here and widely used land cover products revealed a 53% underestimation of wetland extent across the three Arctic sites (Supplementary Fig. 3c), while wetland extent in the four boreal and sub-Arctic sites was approximately twice that of the reference maps (Supplementary Fig. 3c). This suggests that current estimates of wetland $CH_4$ emissions may be overestimated in boreal regions and underestimated in Arctic regions[33,34], particularly at coarse resolutions (Supplementary Fig. 3d). However, these findings are based on a limited number of sites, and broader spatial patterns will require analysis across a larger sample.

Our analysis indicates that mapping resolutions should be near 25 m to limit uncertainty in $CH_4$ emissions to $\sim 13 \pm 9\%$ from mapping-related error in wetland distribution (Fig. 3). Using coarser spatial resolution introduces error related predominately to wetland size but also to wetland type - which can also substantially alter emission estimates (Figs. 3 and 5)[15]. While remote sensing of wetlands has advanced substantially in recent years, global-scale wetland maps are often still produced at relatively coarse spatial resolutions[22,46,47]. In contrast, local and regional maps already reach much finer resolutions with suitable thematic detail but limited spatial extent[21]. Newer global land cover products at 30 m resolution have improved wetland mapping but still lack the thematic detail needed for $CH_4$ budget estimation, particularly for high-latitude wetlands[23,48]. Arctic region datasets at the 10 m resolution can either have similarly limited thematic detail[24] or do not include the Boreal region[49], though the latter represents a significant advancement in Arctic wetland mapping for $FCH_4$ upscaling and provides a strong foundation for extending similar approaches to the broader Arctic-boreal region. Although the flux measurements used here are restricted to the growing season and do not resolve potential variability from seasonality, the growing season typically represents the period of the highest biogeochemical activity and $FCH_4$ in permafrost ecosystems. Consequently, while our recommendations are most directly applicable to the growing season, and we acknowledge the potential for substantial non-growing season emissions[50,51], the contribution of permafrost wetlands to annual $CH_4$ budgets is likely significant and may extend well beyond the growing season.

To improve bottom-up estimates of Arctic and Boreal region $CH_4$ budgets, critical advances are needed: (1) more detailed Arctic-boreal geospatial data ($\leq 25$ m to limit resolution related error to $13 \pm 9\%$) with $FCH_4$-relevant themes (moisture and nutrient regime) and (2) improvements to process-based modeling implementation. Land cover classifications that base wetland classes on hydrological inputs and corresponding nutrient conditions, using vegetation type as a proxy, may more effectively represent distinct $FCH_4$ dynamics across regions with limited spectral variability. Mapping and classifying wetland types with remote sensing requires additional information on soil drainage and peat presence that does not always correlate with current vegetation[52]. Regions with extensive field validation and very high-resolution imagery, as synthesized here, may aid in developing and validating new, higher resolution large-extent wetland products. In regions where wetland extent information is available only at coarse scales, there are limitations in accuracy for global and pan-Arctic and boreal soil and wetland products. This error, associated with coarse wetland representation, can be propagated in wetland data-fusion products and hinders the detailed representation of wetland extent and corresponding $CH_4$ emissions (Fig. 2c), which ideally should be captured at a resolution near 25 m to limit error in $CH_4$ budgets (Fig. 3). Therefore, process-based $CH_4$ models operating on coarser scales (>1 km) should incorporate sub-grid tiling or statistics to account for the extensive distribution of smaller wetland areas across the Arctic and boreal region.

While the highest spatial resolution is desirable, practical constraints necessitate balancing resolution with available resources. Earth observation data from ESA's Sentinel satellite constellations provide higher spatial and temporal resolution compared to single-platform missions such as Landsat. Furthermore, their fully open-access model offers broad accessibility, in contrast to commercial high-resolution constellations (e.g., Maxar, Planet Labs), making Sentinel data particularly valuable for global scientific monitoring of phenological dynamics. Advances in multispectral and radar satellite capabilities have also improved the classification of peatland types (e.g., rich vs. poor fens), moisture gradients, and topographic context. These enhanced sensing capacities allow for more accurate delineation of wetland heterogeneity, particularly in ecotonal zones that are often misrepresented or omitted in lower-resolution products. However, it is important to recognize that wetlands and peatlands remain challenging to classify, as their land cover types and vegetation communities are often highly patterned and fragmented[26], and their spectral properties vary seasonally with changes in moisture conditions and vegetation phenology[53]. Nevertheless, processing capabilities are advancing rapidly, enabling the use of adequate resolution remote sensing datasets for global-scale land cover mapping[54,55], including applications for monitoring global surface water dynamics and mapping wetland extents[56]. New directions include machine learning-based upscaling techniques for land cover detection or change trend analysis[57,58] and AI-based Foundation Models[59] that will increasingly enable rapid analysis of high-resolution, near-real time observation data from highly dynamic environments such as wetlands. Our study underscores the importance of utilizing spatially and thematically high-resolution wetland maps to capture Arctic and boreal $FCH_4$ at sufficient detail, and with lower uncertainty, to inform carbon cycle models. The important discussion of to what extent the Arctic and boreal region C sink and $CH_4$ source contribute to the global C cycle and how they change in a rapidly warming Arctic[4,10] builds on our ability to successfully map and classify wetland distribution and type. In addition to mapping these remote areas, additional aircraft campaigns focused on atmospheric $CH_4$ measurements can help to validate the spatial distribution of $CH_4$ emissions[60-62].

## Conclusions

High-resolution wetland extent and category data are critical for accurately estimating $CH_4$ emissions in northern high latitudes. A spatial resolution near or finer than 25 m for global or pan-Arctic and boreal products should be used, although some regions with high landscape heterogeneity may need even finer spatial resolutions. While techniques involving aggregation to coarser resolutions while reporting subgrid land cover fractions offer methods for reducing input parameter size while retaining representativity, they rely on accurate underlying data that must be at the proper spatial resolution needed to describe land cover morphology. Similarly, spectral unmixing techniques on coarser products that provide fractional coverage of land cover type within a pixel must be able to provide fractional estimates

https://doi.org/10.1038/s43247-025-02963-1  **Article**

that match that of higher resolution products while also differentiating similar land cover types that are biogeochemically relevant for $CH_4$ (i.e., fens, bogs and other wetland types).

There are numerous thematic peatland and wetland classification systems, each employing a variety of methods for interpreting remotely sensed data. It is essential to clearly describe the classification criteria used, as there is a clear need for greater standardization in this field. Several current Earth observation datasets support the recommended resolution (e.g. Landsat, Sentinel) but must be supplemented with additional geospatial information to identify wetland types (e.g., bogs, fens) crucial for $FCH_4$ processes. Estimates of $FCH_4$ derived solely from surface inundation or generic wetland classes may overlook key landscape features, leading to significant errors and uncertainty. Despite the growing number of regional and global bottom-up $FCH_4$ estimates, caution must be exercised regarding persistent biases arising from poor spatial representation, artificial redistribution of wetlands in coarse-resolution datasets, and inadequate classification of wetland types. Continued coordinated research efforts have strong potential to leverage advances in computing and Earth observation technologies to improve understanding of $CH_4$ budgets in high-latitude ecosystems.

## Methods
### Data selection
To conduct this synthesis study, we identified existing land cover classification datasets encompassing Arctic and boreal region wetlands (Fig. 1). These datasets comprised published studies and/or available datasets with very high spatial resolution land cover classification maps[14,35–37,63–66] (<2.5 m) and where chamber-based $FCH_4$ measurements were conducted[14,30,35–37,67–69] that represented the array of land cover classes in the land cover maps (Supplementary Table 1). Datasets were chosen based on spatial and temporal coverage, data quality, and alignment with our research objectives. Specifically, selected land cover classification maps needed to possess spatial resolutions of less than 2.5 m, cover areas of at least 2.5 km², and have been generated within five years of relevant $FCH_4$ measurements. The land cover maps are described in more detail in Supplementary Table 1 as well as in the primary sources. Due to the large spatial extent of the mapped area of the Scotty Creek watershed (~ 12 × 19 km), this region was split into northern and southern locations, North Scotty Creek and South Scotty Creek, respectively.

### Land cover class harmonization
The thematic detail of the reference maps varied significantly, ranging from low resolutions where wetland types were simply labeled as 'wetland' to high-resolution products detailing various wetland classes based on factors like moisture gradient, nutrient availability, and successional stages in wetland development (Supplementary Table 1). This variability in thematic detail directly impacts pixel patch size, as larger objects are more likely to be represented at coarser scales. To address potential scaling issues, land cover types were aggregated and harmonized across maps, reducing the number of classes to 5–7 (Supplementary Table 1). The general categories included upland/grassy, barren, forest, wetlands, and lakes. We tested the sensitivity to distinguishing different wetland classes by combining fen and bog classes at higher-resolution land cover themes and repeating the scaling analysis. For the Utqiaġvik area, the land cover map was aggregated into 'permafrost bog' and 'permafrost wetland' categories, aligning with the BAWLD classification theme characteristic of this continuous permafrost wetland region[3,13]. Lakes and rivers were not included in this analysis due to limited representation in $FCH_4$ measurements within our datasets.

### Resampling
To investigate the impact of spatial resolution on $CH_4$ emission estimates, emissions derived from low spatial resolution land cover maps were compared to those derived from the reference maps (original high resolutions). Using QGIS software[70], we generated coarser land cover classifications by mode-aggregating the reference maps from the base resolutions to coarser

resolutions at 5, 10, 25, 50, 100, 250, 500, 1000, 1250, and 5000 meter steps. Mode aggregation assigns a pixel as the dominant (majority) land cover class within the pixel extent, replicating how many land cover products are constructed. This approach allowed us to coarsen the high-resolution reference maps in a way that is representative of many existing products. The coarsened resolutions were specifically selected to align precisely with the spatial extent of the maps. Each resolution step ensures that no partial pixels are left at the map edges. This approach preserves spatial integrity and avoids artifacts introduced by edge truncation, allowing a consistent grid structure across all scales. As such, we would compare 5, 10, 25, and 1000 m resolutions to SPOT, Sentinel, Landsat, and MODIS image resolutions respectively. $CH_4$ emissions presented here were calculated based on the progressively coarser land cover map to analyze the conservation of $CH_4$ emissions magnitudes at different spatial resolutions and identify the threshold at which significant deviations from the reference maps occurred. Prior to resampling, each land cover map was cropped to ensure alignment compatibility with the chosen resolution steps, eliminating any potential overhang that could affect emission estimates. It is important to acknowledge that the spatial resolutions and boundary definitions in the reference maps at nominal resolution could influence our $CH_4$ emission estimates and the subsequent analysis.

### $FCH_4$ measurements and upscaling
Methane fluxes were measured and averaged for each land cover class at each site (Supplementary Table 2) using the manual chamber technique. The number of plots per site ranged from 12 to 279 with chamber areas between 0.03 to 0.36 m² and were taken during the growing seasons from 2007 to 2019. Specific methodologies and measurement frequencies for fluxes and land cover maps are described in the primary sources (Supplementary Table 1). Negative $FCH_4$ values indicate uptake while positive values indicate emission. The high-resolution land cover maps were used to upscale $FCH_4$ across the various regions. To calculate regional $FCH_4$, we used an upscaling approach:

$$FCH4_{region} = \sum_{1}^{n}(A_i \times FCH4_i) \div A_{region}$$

where $i$ represents the individual land-cover classes within each region, $n$ represents the total number of land cover classes (Supplementary Table 1), $A$ represents the area and $FCH_4$ represents the mean of the measured $FCH_4$ for the land cover class. This methodology, akin to a "paint-by-numbers" approach, simplifies the estimation of regional emissions, offering only a snapshot of emissions at a specific point in time. This method does not capture temporal variability or provide mechanistic insights into $FCH_4$ and $CH_4$ budgets but instead establishes a framework for evaluating resolution requirements for accurately upscaling $CH_4$ emissions. This approach assumes that each land cover class has a representative flux and does not capture fine-scale hydrological or microtopographic heterogeneity within classes. Fluxes for inland waters, which comprised an average of 1.7% of the land area, were not typically collected during in-situ campaigns. Instead, we used values from the BAWLD-$CH_4$ database[3] for "medium-sized peatland lakes", where the median total flux—summing diffusive and ebullitive contributions—was 63.5 mg $CH_4$ m$^{-2}$ d$^{-1}$ (diffusion: 18.4; ebullition: 45.1). To assess variability, we examined the 25$^{th}$ and 75$^{th}$ percentile estimates, which were 31.8 mg $CH_4$ m$^{-2}$ d$^{-1}$ (diffusion: 11; ebullition: 20.8) and 122.5 mg $CH_4$ m$^{-1}$ d$^{-1}$ (diffusion: 42; ebullition: 80.5), respectively. This variability had little impact on the deviation from the regional reference flux estimates (Supplementary Fig. 4).

### Statistical analysis
Regressions and plots were generated in R software (v4.3.3)[71] using the 'ggplot2'[72], 'ggsignif'[73], 'data.table'[74] packages. Percent deviation from reference (Fig. 3; Supplementary Fig. 2) is the absolute value of the difference of regional $FCH_4$ at the nominal resolution and regional $FCH_4$ at a spatial resolution step divided by the regional $FCH_4$ at the nominal resolution.

Landscape division index and patch statistics (Fig. 4) were computed using FragStats software[75]. Resolution sensitivity scores (Fig. 4) indicate the relative sensitivity of a region's $FCH_4$ signal to spatial resolution coarsening, ranked from 1 (most sensitive) to 7 (least). Scores were derived as the area under the deviation-from-reference curve for each region (Supplementary Fig. 2). Areas were calculated using the trapezoidal rule over log-transformed resolution steps to normalize distance between resolutions.

## Data availability

Data used in this study are available via the following sources: WAD2M wetland data (Fig. 1) is available at https://doi.org/10.5281/zenodo.3998454. Land cover classification maps and data pertaining to resolution resampling is available at https://doi.org/10.5281/zenodo.17085658.

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

## Acknowledgements
This work is part of the NASA-ESA Arctic Methane and Permafrost Challenge (AMPAC-Net project 4000137912/22/I-DT) and partially funded by European Research Council (ERC) Starting Grant FluxWIN (No. 851181). Data from Pallas, FI was collected with the Academy of Finland funding awarded to Tarmo Virtanen (No. 308513 & 349193). Chamber fluxes from Utqiaġvik were collected with funding from the Office of Polar Programs of the National Science Foundation awarded to D. Zona, W.C. Oechel (No. 1932900, 2149988) with additional support by NOAA CESSRST (No. NA16SEC4810008) to W.C. Oechel. We thank Craig E. Tweedie and the Barrow Area Information Database (BAID) for the land cover classification from Utqiaġvik, AK, US. Finally, we would like to thank the reviewers for their time and effort in providing valuable feedback, helping to improve this study.

## Author contributions

C.C. Treat, G. Grosse and J. Hashemi conceived the work. A. Räsänen and J. Hashemi conducted the analysis. J. Hashemi generated the visualizations. J. Hashemi wrote the manuscript in close collaboration with C. C. Treat. C.C. Treat, A. Räsänen, T.Virtanen, S. Juutinen, L. Chasmer, A.M. Virkkala, C. Voigt, O. Sonnentag, M. Korkiakoski, M. Aurela, M. Luoto, P. Niittynen, and S.J. Davidson provided land cover classification maps and/or in-situ chamber flux data. J. Hashemi, A. Räsänen, T. Virtanen, S. Juutinen, G. Grosse, M. Aurela, A. Bartsch, L. Chasmer, S. J. Davidson, M. Korkiakoski, M. A. Kuhn, M. J. Lara, M. Luoto, P. Niittynen, D. Olefeldt, O. Sonnentag, A. M. Virkkala, C. Voigt, C. C. Treat contributed to the interpretation of data and writing.

## Funding

## Competing interests

The authors declare no competing interests.

## Additional information

[1]Permafrost Research Section, Alfred Wegener Institute, Helmholtz Centre for Polar and Marine Research, Potsdam, Germany. [2]Geography Research Unit, University of Oulu, Oulu, Finland. [3]Ecosystems and Environment Research Programme, University of Helsinki, Helsinki, Finland. [4]Finnish Meteorological Institute, Climate System Research, Helsinki, Finland. [5]Institute of Geosciences, University of Potsdam, Potsdam, Germany. [6]b.geos GmbH, Korneuburg, Austria. [7]Department of Geography and Environment, University of Lethbridge, Lethbridge, AB, Canada. [8]School of Geography, Earth and Environmental Sciences, University of Plymouth, Plymouth, United Kingdom. [9]Département des sciences biologiques, Université du Québec à Montréal, Montréal, QC, Canada. [10]Department of Geography, University of British Columbia, Vancouver, BC, Canada. [11]Department of Geography/Plant Biology, University of Illinois, Urbana, IL, USA. [12]Department of Geosciences and Geography, University of Helsinki, Helsinki, Finland. [13]Department of Environmental Sciences, University of Jyväskylä, Jyväskylä, Finland. [14]Department of Renewable Resources, University of Alberta, Edmonton, AB, Canada. [15]Département de géographie, Université de Montréal, Montréal, QC, Canada. [16]Woodwell Climate Research Centre, Falmouth, MA, USA. [17]Institute of Soil Science, Universität Hamburg, Hamburg, Germany. [18]Department of Agroecology, Aarhus University, Aarhus, Denmark. ✉e-mail: joshua.hashemi@awi.de

