## [Transparent Peer Review file · Communications Earth & Environment]

Coarse land cover datasets bias Arctic-boreal wetland methane budgets

Corresponding Author: Dr Josh Hashemi

Version 0:

Decision Letter:

Dear Dr Hashemi,

Your manuscript titled "Coarse land cover datasets bias Arctic-boreal wetland methane budgets" has now been seen by 3 reviewers, and we include their comments at the end of this message. They find your work of interest, but some important points are raised. We are interested in the possibility of publishing your study in Communications Earth & Environment, but would like to consider your responses to these concerns and assess a revised manuscript before we make a final decision on publication.

We therefore invite you to revise and resubmit your manuscript, along with a point-by-point response that takes into account the points raised. Please provide a thorough analysis of associated uncertainties, such as from FCH₄ data, and the assumption of 0 CH₄ flux for the 'water' land cover class. Please highlight all changes in the manuscript text file.

Please submit your point-by-point responses as a separate file, distinct from your cover letter where you can add responses to the Editors' comments that you do not want to be made available to the reviewers. Word files are preferred. We recommend that any figures, tables or graphs that are included in the response to reviewers are also included in the main article or Supplementary Information.

Please use the following link to submit your revised manuscript, point-by-point response to the referees' comments (which should be in a separate document to any cover letter), a tracked-changes version of the manuscript (as a PDF file) and the completed checklist:

Link Redacted

We hope to receive your revised paper within six weeks; please let us know if you aren't able to submit it within this time so that we can discuss how best to proceed. If we don't hear from you, and the revision process takes significantly longer, we may close your file. In this event, we will still be happy to reconsider your paper at a later date, as long as nothing similar has been accepted for publication at Communications Earth & Environment or published elsewhere in the meantime.

Please do not hesitate to contact us if you have any questions or would like to discuss these revisions further. We look forward to seeing the revised manuscript and thank you for the opportunity to review your work.

Best regards,

Mengze Li, PhD
Editorial Board Member
Communications Earth & Environment

Somaparna Ghosh, PhD
Associate Editor - Communications Earth & Environment
Consulting Editor - Communications Sustainability

EDITORIAL POLICIES AND FORMATTING

- Behavioural and social science
- Ecological, evolutionary & environmental sciences
- Life sciences

Furthermore, please align your manuscript with our format requirements, which are summarized on the following checklist: <https://www.nature.com/documents/commsj-phys-style-formatting-checklist-article.pdf> Communications Earth & Environment formatting checklist

and also in our style and formatting guide <https://www.nature.com/documents/commsj-phys-style-formatting-guide-accept.pdf> Communications Earth & Environment formatting guide .

*** DATA: Communications Earth & Environment endorses the principles of the Enabling FAIR data project (<http://www.copdess.org/enabling-fair-data-project/>). We ask authors to make the data that support their conclusions available in permanent, publically accessible data repositories. (Please contact the editor if you are unable to make your data available).

All Communications Earth & Environment manuscripts must include a section titled "Data Availability" at the end of the Methods section or main text (if no Methods). More information on this policy, is available at <http://www.nature.com/authors/policies/data/data-availability-statements-data-citations.pdf>.

If a community resource is unavailable, data can be submitted to generalist repositories such as <https://figshare.com/> or <http://datadryad.org/> Dryad Digital Repository. Please provide a unique identifier for the data (for example a DOI or a permanent URL) in the data availability statement, if possible. If the repository does not provide identifiers, we encourage authors to supply the search terms that will return the data. For data that have been obtained from publically available sources, please provide a URL and the specific data product name in the data availability statement. Data with a DOI should be further cited in the methods reference section.

REVIEWER COMMENTS:

Reviewer #1 (Remarks to the Author):

This is a compelling manuscript that addresses an important gap in our understanding of the global methane budget – how to best translate field measurements to larger regional scale estimates using remote sensing data. The study demonstrates how regional flux estimates across 7 wetland sites in the Arctic-Boreal region are affected by the spatial resolution of land cover data, by estimating methane flux over each study site using progressively coarsened land cover maps from 2.5 m resolution up to 5 km. Insights are provided regarding the influence of landscape structure on how sensitive flux estimates are to map coarsening, and a clear recommendation is made to minimize error in flux estimates by using products with 25 m

spatial resolution or finer. There are places where the text and figures could be slightly improved enhance clarity, and a few topics that could be addressed to further enhance the insights provided in the manuscript (variability in the methane flux measurements used in the model, the aggregation method used for coarsening). Overall, the paper clearly presents a relatively well-designed study with compelling results and important implications. My comments and suggestions are detailed below:

Main Comments:

1. It is unclear how variability in methane flux rates was considered. How variable were flux rates in the underlying data sets and how were flux values selected/calculated (e.g. mean of all values?) While it is clearly stated that the analysis does not capture temporal variability and is only a snapshot view of emissions (Line 358-359), ignoring seasonal fluctuations can result in substantial bias when creating inventory-based methane emission estimates, especially when using high spatial resolution data (Hondula et al. 2021; Kyzivat and Smith 2023). Can the authors comment on the potential magnitude of the influence of flux rate variability on emission estimates, compared to the differences based on spatial resolutions alone? Would the authors have any recommendations or insights based on their analysis about how to best leverage higher spatial resolution data for estimating annual emissions in Arctic-Boreal regions?
2. Although references to original sources of the methane flux data are provided in Table S1, having an additional supplemental table providing the methane flux estimates (ie. the data included as “Methane flux estimates for each of the reference maps used at different resolutions”) would be very valuable (or even figure in the main manuscript). Having the key details (methane flux rates) underlying the present study be more accessible would greatly enhance the manuscript. The appropriate number of significant digits of the FCH₄ measurements should also be checked since the values suggest an unusually high degree of precision.
3. It is somewhat confusing to assign 0 emissions as the flux estimate for the “water” land cover class, given the high methane emissions reported from systems like permafrost thaw ponds and concern about their contribution to climate feedbacks (e.g. Kuhn et al. 2018; Prékienis et al. 2021; Laurion et al. 2009). The methods (Line 361-365) explain that emissions from inland waters (rivers and lakes) are ignored because of limited flux measurements from those land cover types, but this assumption is inconsistent with the results often being presented as “regional FCH₄ emissions”. I trust the authors judgment on the reconciliation of various land cover classes being as consistent as possible across study areas, but the exclusion of water areas is highly consequential for the implications of the study because of the persistent confusion attributed to wetland and open water classes in wetland emission estimates (i.e. the so-called “double counting” problem, Thornton et al. 2016). I would encourage the authors to consider testing the sensitivity of their results to including a non-zero flux estimate for water (e.g. using a ballpark best guess high and low estimate from the literature) – I suspect it would underscore the main patterns in their findings, but could substantially influence the error magnitudes as reported in Figure 3. Otherwise, the discussion and conclusions should justify and note this assumption much more prominently and comment on its implications.
4. Line 329: It would strengthen the manuscript to provide some more insight and justification about the implications of using mode aggregation to coarsen the spatial resolution. How do the results of mode aggregation compare to the spatial patterns for each site in the datasets included in Supplementary Figure 3? Since many of the coarser resolution wetland data sets report fractional cover, and other metrics that can be derived from spectral unmixing, it seems like it could be a more realistic comparison to represent fractional coverage in the coarsened data. A figure showing how the spatial patterns in the Figure S3 datasets compare to the coarsened estimates shown in Figure S1 would be very insightful in this regard. Figure S3 is highly insightful and would be of great interest to users of these datasets – consider moving to the main text and adding points on top of the boxplots to show the values for each of the study sites.
5. There are some places where the language could be easily mis-interpreted – consider revising to be more precise and avoid confusion:
 - a. Line 118: Consider rephrasing to clarify that this relationship is between estimated/modeled emissions and wetland area since neither variable was “observed” at the regional scale.
 - b. Line 126: “change in wetland extent” – consider rephrasing, e.g. “difference in estimated wetland extent” to avoid sounding like a real world change over time.
 - c. Line 152: “first to exhibit” – consider rephrasing to avoid sounding like a change over time, for example something like “For example, CH₄ estimates from the Kilpisjarvi region transitioned from source to sink at the finest spatial resolution”
 - d. Line 217: “emissions” should be “emission estimates”

Minor comments

6. Line 51: Is “potential” necessary here before “emissions”?
7. Line 52: Should “requires” here be “require”?
8. Line 55: Please specify uncertainty in what, perhaps “... contribute to the overall uncertainty of CH₄ emission estimates for regions...”
9. Line 56: Remove comma after differences in “for differences, in CH₄ flux ... “
10. Line 67-75: Consider specifying, even in parentheticals, some of the relevant spatial resolutions or sizes of features mentioned in this section, i.e. how narrow do you mean by “narrow channels” and what are “these resolutions” that “may still be too coarse”.
11. Line 73-75: I’m slightly confused by how the two issues are related in this sentence – I agree both are issues, but wetland-upland boundary mapping seems to be about intra wetland variability from different zones/patches within a wetland site, whereas observational bias towards high-emitting wetlands is at the wetland site scale.
12. Line 83-86: It may help the reader here to distinguish between spaceborne and airborne sensors, e.g. “... from spaceborne sensors like MODIS... are freely available on a global scale, very high spatial resolution data (<2.5m) available from airborne sensors is often...”.
13. Line 101: Should “variability in spatial resolution” here be just “spatial resolution” or “differences in spatial resolution”? As written this could be interpreted as comparing maps with low variability in spatial resolution to maps with high variability

in spatial resolution, and its unclear what that would mean.

14. Line 141: What does “near” mean in this sentence? Consider rephrasing as “similar to” and/or mention the specific datasets being referred to here.

15. Line 214: Is there a word missing here? “...should near 25 m”

16. Figure 1: Treeline boundary line is similar color to the high end of the color palette for wetland distribution. Is the treeline necessary to include? If so, consider using a color more distinct than the wetland distribution to avoid confusion. The abbreviations (TP, HBL, etc.) also aren't referenced elsewhere in the paper, are they necessary to include?

17. Supplementary Table 1: If they are notably different, consider including a column with a brief indication of what methods/type of data each source is based on (e.g. multi-spectral vs hyperspectral, lidar, etc.; any notable differences in classification techniques)

18. How does the highest spatial resolution (2.5 m) resolution compare to the area of chamber measurements?

19. Supplementary Figure 2: What unit is area reported in?

20. Figure 2b – it is unclear what the error/uncertainty is in this figure. Is this across spatial resolutions?

21. It would be interesting to see how the pattern shown in Figure 3 compares across sites. Is it possible to include additional lines (different colors?) showing how the sites vary?

22. Figure 4: The y-axis scale is somewhat counterintuitive, in that the more sensitive sites have lower sensitivity scores. Consider reversing this axis or perhaps re-naming the y-axis to “rank” instead of “score”.

23. Supplementary Figure 1: This is a very nice figure, however the legends in the bottom row are not readable, at least when printed out. It would be nice to be able to see what classes are included for each site. Is it possible to increase the text size, or separate the legends into a row below the current panels?

24. Please provide references/more details for the datasets used in Supplemental Figure 3. It would be very useful to also provide the spatial resolution of each, to be able to better link to the study results.

References

Hondula, Kelly L., et al. "Effects of using high resolution satellite-based inundation time series to estimate methane fluxes from forested wetlands." *Geophysical Research Letters* 48.6 (2021): e2021GL092556.

Kuhn, McKenzie, et al. "Emissions from thaw ponds largely offset the carbon sink of northern permafrost wetlands." *Scientific Reports* 8.1 (2018): 9535.

Kyzivat, Ethan D., and Laurence C. Smith. "A closer look at the effects of lake area, aquatic vegetation, and double-counted wetlands on Pan-Arctic Lake methane emissions estimates." *Geophysical Research Letters* 50.24 (2023): e2023GL104825.

Laurion, Isabelle, et al. "Variability in greenhouse gas emissions from permafrost thaw ponds." *Limnology and Oceanography* 55.1 (2010): 115-133.

Prėskienis, Vilmantas, et al. "Seasonal patterns in greenhouse gas emissions from lakes and ponds in a High Arctic polygonal landscape." *Limnology and Oceanography* 66 (2021): S117-S141.

Thornton, Brett F., Martin Wik, and Patrick M. Crill. "Double-counting challenges the accuracy of high-latitude methane inventories." *Geophysical Research Letters* 43.24 (2016): 12-569.

Reviewer #2 (Remarks to the Author):

The manuscript titled " Coarse land cover datasets bias Arctic-Boreal wetland methane budgets" was a unique and novel study. This study provides crucial empirical evidence demonstrating the necessity of high-resolution (≤ 25 m) land cover data with appropriate thematic detail for accurately estimating CH₄ emissions in Arctic-boreal wetland. The findings show that coarse-resolution (> 1 km) maps misrepresent wetland extent and type—particularly high-emitting fens—leading to erroneous source/sink conclusions. This work is highly valuable for reducing global methane budget uncertainties and sets clear benchmarks for future upscaling efforts. Some specific comments are following:

LN111: Unclear with “at all sites and at four sites”, which four sites?

LN211: “33.34.”? Please provide correct citation format.

LN583: Could you provide another table to show the CH₄ flux of each land cover type of the 7 sites you collected from other studies? It could be a supplementary table to support the result of Figure 2c and Supplementary Figure 1 to see how the regional CH₄ fluxes changed with the alternation of land cover types.

LN589/Figure 5: Is the contribution simply based on the area of bog and fen in each spatial resolution class? According to Fig. 2a, the wetland area (e.g. fen and bog) tends to be zero when spatial resolution is larger than 1 km, so CH₄ emissions from wetlands would also be zero. How did you calculate the contribution of fen and bog to FCH₄ at coarser resolution?

Supplementary Figure 1: The legends in the figure were unreadable. In addition. It would be better to uniform the color of each land cover type across 7 sites. Moreover, lowest spatial resolution here was 1 km, why didn't you show the results of 1250, 2500 and 5000 m so that they were consistent with other figures and tables?

Supplementary Table 1: Please uniform the font size below the heading line.

Reviewer #3 (Remarks to the Author):

I thought this was an elegantly put together paper that made a specific point and provided the proper analyses to show what effect the coarseness of remote sensing has on estimated emissions from regional CH₄ fluxes. It makes a compelling case that coarse resolution estimates of the regional flux are almost certainly introducing substantial error.

The conclusions seem sound basically everywhere from my interpretation. There are some limits to what the paper can state realistically, but the authors do a good job recognizing those and pointing them out. I also don't think there's an easy way around those limitations.

I wouldn't say the results are all that surprising per se, but I'm not aware of anyone who has done this analysis before, and I think it is a useful paper that will be cited frequently. This is a big issue that the field is having to tackle.

I have really just a few minor nitpicks that I thought might result in a slightly higher quality presentation.

358-365 - I appreciated the caveats here. One additional caveat that I think should be listed is heterogeneity within any individual wetland (e.g. hydrological variation within the wetland, edge vs center of a particular terrain type, etc). I don't think it affects your analysis at all but it was something I kept thinking about as I viewed your maps.

Figure 3. Caption should make clearer what shaded area is

Figure 4. You could eliminate y labels and ticks on this graph if the y limits of the panels were lightly adjusted to match those of c

Supplementary Table 1 - It took me a few looks to figure out how to read this table. Could the caption be edited to make clear you need to refer to the supplementary bibliography just to make it more clear? I was looking for footnotes and such at first

Supplementary Figure 1 - The text in the legend is so small it is unreadable in all the files provided for review. I assume there's a vector art image that could be used that would address this. It's a small problem, but should be addressed

** Visit Nature Portfolio's author and referees' website at www.nature.com/authors for information about policies, services and author benefits**

Communications Earth & Environment is committed to improving transparency in authorship. As part of our efforts in this direction, we are now requesting that all authors identified as 'corresponding author' create and link their Open Researcher and Contributor Identifier (ORCID) with their account on the Manuscript Tracking System prior to acceptance. ORCID helps the scientific community achieve unambiguous attribution of all scholarly contributions. You can create and link your ORCID from the home page of the Manuscript Tracking System by clicking on 'Modify my Springer Nature account' and following the instructions in the link below. Please also inform all co-authors that they can add their ORCIDs to their accounts and that they must do so prior to acceptance.
<https://www.springernature.com/gp/researchers/orcid/orcid-for-nature-research>

If you experience problems in linking your ORCID, please contact the Platform Support Helpdesk.

Version 1:

Decision Letter:

Dear Dr Hashemi,

Your manuscript titled "Coarse land cover datasets bias Arctic-boreal wetland methane budgets" has now been seen by our reviewers, whose comments appear below. In light of their advice we are delighted to say that we are happy, in principle, to publish a suitably revised version in Communications Earth & Environment.

We therefore invite you to revise your paper one last time to address the remaining concerns of our reviewers. At the same time we ask that you edit your manuscript to comply with our format requirements and to maximise the accessibility and therefore the impact of your work.

EDITORIAL REQUESTS:

*****Please take care to match our formatting and policy requirements. We will check revised manuscript and return manuscripts that do not comply. Such requests will lead to delays. *****

SUBMISSION INFORMATION:

OPEN ACCESS:

Communications Earth & Environment is a fully open access journal. Articles are made freely accessible on publication. For further information about article processing charges, open access funding, and advice and support from Nature Portfolio, please visit <https://www.nature.com/commsenv/open-access>

Link Redacted

Best regards,

Somaparna Ghosh, PhD
Associate Editor,
Communications Earth & Environment
Consulting Editor,
Communications Sustainability

REVIEWERS' COMMENTS:

Reviewer #1 (Remarks to the Author):

The authors have done a commendable job improving the manuscript to address the concerns, and the updates and expanded analyses underscore the main findings making this a more compelling study overall and I look forward to seeing it published.

One very small remaining item that could improve clarity is to specify what the points indicate for the boxplots in supplementary figure 3.

Also, for lines 361-363, this statement would be stronger with a supporting reference and/or mention of specific land cover products.

Reviewer #2 (Remarks to the Author):

The author has satisfactorily addressed the issues I raised.

** Visit Nature Portfolio's author and referees' website at www.nature.com/authors for information about policies, services and author benefits**

Dear Reviewers,

Thank you for the time and effort you devoted to reviewing our manuscript. We have incorporated literature values for lake emissions from the BAWLD-CH₄ database, added flux values and associated uncertainties for each land-cover class at each study site, corrected one flux value for the Tiksi site, and clarified the rationale for our aggregation method. We have also expanded the Supplementary Information to include comparisons with coarsened spatial resolution data with existing land-cover products. In addition, we implemented the recommended minor revisions and updated the figures and tables accordingly. We feel that your input and the corresponding revisions have resulted in a substantial improvement to this work. Please find our responses to each of your comments below.

REVIEWER COMMENTS:

Reviewer #1 (Remarks to the Author):

This is a compelling manuscript that addresses an important gap in our understanding of the global methane budget – how to best translate field measurements to larger regional scale estimates using remote sensing data. The study demonstrates how regional flux estimates across 7 wetland sites in the Arctic-Boreal region are affected by the spatial resolution of land cover data, by estimating methane flux over each study site using progressively coarsened land cover maps from 2.5 m resolution up to 5 km. Insights are provided regarding the influence of landscape structure on how sensitive flux estimates are to map coarsening, and a clear recommendation is made to minimize error in flux estimates by using products with 25 m spatial resolution or finer. There are places where the text and figures could be slightly improved enhance clarity, and a few topics that could be addressed to further enhance the insights provided in the manuscript (variability in the methane flux measurements used in the model, the aggregation method used for coarsening). Overall, the paper clearly presents a relatively well-designed study with compelling results and important implications. My comments and suggestions are detailed below:

Main Comments:

1. It is unclear how variability in methane flux rates was considered. How variable were flux rates in the underlying data sets and how were flux values selected/calculated (e.g. mean of all values?) While it is clearly stated that the analysis does not capture temporal variability and is only a snapshot view of emissions (Line 358-359), ignoring seasonal fluctuations can result in substantial bias when creating inventory-based methane emission estimates, especially when using high spatial resolution data (Hondula et al. 2021; Kyzivat and Smith 2023). Can the authors comment on the potential magnitude of the influence of flux rate variability on emission estimates, compared to the differences based on spatial resolutions alone? Would the authors have any recommendations or insights based on their analysis about how to best leverage higher spatial resolution data for estimating annual emissions in Arctic-Boreal regions?

Thank you for the comment. The flux measurements were means per land cover class - this has been added to the methods (L378 & 389). The uncertainty (standard error) for the flux measurements has been added to the underlying dataset, Supplementary Table 2, and Figure 2c and throughout the manuscript text, with specific attention to the variability of flux uncertainty vs error from spatial resolution (L153-155).

There are unfortunately not enough data from the chamber-based measurements here to come to annual budgets that can account for how these results would change due to seasonality. Eddy covariance datasets provide more insight on seasonality, yet these measurements integrate area over coarser scales than the resolutions used in the reference land cover maps and would require footprint modelling/downscaling to the land cover class level, introducing more uncertainty to the analysis. Though, as the growing season represents the period of the highest microbial activity, we feel that the recommendations provided hold when extending to annual budgets. We have added some discussion on this point to the manuscript (L242-248).

2. Although references to original sources of the methane flux data are provided in Table S1, having an additional supplemental table providing the methane flux estimates (i.e. the data included as “Methane flux estimates for each of the reference maps used at different resolutions”) would be very valuable (or even figure in the main manuscript). Having the key details (methane flux rates) underlying the present study be more accessible would greatly enhance the manuscript. The appropriate number of significant digits of the FCH₄ measurements should also be checked since the values suggest an unusually high degree of precision.

Thank you for this suggestion. We have added a supplementary table (Supp. Table 2) reporting flux rates for each land cover class at each site. Figure 2c shows flux rates as they change with resolution. Including all land cover types for each map across all resampled resolutions would essentially reproduce the dataset, which is publicly available at the DOI provided in the Data Availability section. Regarding the number of digits in the underlying data, fluxes have been rounded, though we note that these values are not unusual for flux measurements, as it reflects the digits carried forward from the original concentration-change slopes involved in flux calculations, averaging across replicates, and unit conversions. Similar values are reported in literature datasets (e.g., Virkkala et al., 2021).

Virkkala, A.-M. et al. (2021). The ABCflux Database: Arctic-Boreal CO₂ Flux and Site Environmental Data, 1989-2020 (Version 1). ORNL Distributed Active Archive Center. <https://doi.org/10.3334/ORNLDAAC/1934>

3. It is somewhat confusing to assign 0 emissions as the flux estimate for the “water” land cover class, given the high methane emissions reported from systems like permafrost thaw ponds and concern about their contribution to climate feedbacks (e.g. Kuhn et al. 2018; Prèskienis et al. 2021; Laurion et al. 2009). The methods (Line 361-365) explain that emissions from inland waters (rivers and lakes) are ignored because of limited flux measurements from those land

cover types, but this assumption is inconsistent with the results often being presented as “regional FCH₄ emissions”. I trust the authors judgment on the reconciliation of various land cover classes being as consistent as possible across study areas, but the exclusion of water areas is highly consequential for the implications of the study because of the persistent confusion attributed to wetland and open water classes in wetland emission estimates (i.e. the so-called “double counting” problem, Thornton et al. 2016). I would encourage the authors to consider testing the sensitivity of their results to including a non-zero flux estimate for water (e.g. using a ballpark best guess high and low estimate from the literature) – I suspect it would underscore the main patterns in their findings, but could substantially influence the error magnitudes as reported in Figure 3. Otherwise, the discussion and conclusions should justify and note this assumption much more prominently and comment on its implications.

Thank you for the suggestion. We have included flux values for inland waters from the BAWLD-CH₄ database (Kuhn et al., 2021) corresponding to the 25% and 75% percentile estimates of the sum of diffusive (1st & 3rd Quartile: 11 & 42 mg CH₄ m⁻² d⁻¹) and ebullitive (20.8 & 80.5 mg CH₄ m⁻² d⁻¹) CH₄ fluxes from medium sized peatland lakes. Because there is very little variability between the low and high estimates in the deviation from reference curves (Fig. A), we have now used the median value (Fig. B) in the manuscript and included Fig. A in the supplementary (Supplementary Figure 4). We have updated all figures and revised the text in the manuscript to reflect the adjusted inland water flux values. (L202-205; L395-403; Figures 2, 3, 4 & 5; Supplementary Figures 2 & 4; Supplementary Table 2)

Kuhn, M. A. et al. 2021. BAWLD-CH₄: a comprehensive dataset of methane fluxes from boreal and arctic ecosystems. *Earth Syst. Sci. Data* 13, 5151–5189.

4. Line 329: It would strengthen the manuscript to provide some more insight and justification about the implications of using mode aggregation to coarsen the spatial resolution. How do the results of mode aggregation compare to the spatial patterns for each site in the datasets included in Supplementary Figure 3? Since many of the coarser resolution wetland data sets report fractional cover, and other metrics that can be derived from spectral unmixing, it seems like it could be a more realistic comparison to represent fractional coverage in the coarsened data. A figure showing how the spatial patterns in the Figure S3 datasets compare to the coarsened estimates shown in Figure S1 would be very insightful in this regard. Figure S3 is highly insightful and would be of great interest to users of these datasets – consider moving to the main text and adding points on top of the boxplots to show the values for each of the study sites.

We appreciate the feedback. We used mode aggregation because it reflects how many coarse-resolution land cover products are constructed, with each pixel assigned the dominant land cover type within its extent. This approach allowed us to coarsen our high-resolution reference maps in a way that is representative of the structure of many existing products not reporting fractional coverage. We agree that land cover products that use fractional cover can provide increased accuracy of proportions of land cover types within a pixel – given that the spectral signals are not too similar (as is often the case for wetland regions in differentiating wetland type). However, if our analyses were repeated using preserved fractional land cover from the highest resolution, the upscaling would not introduce any differences and we would not expect any change in the methane signal. Our use of mode aggregation was therefore intended to illustrate the mapping related error originating from coarse landcover classifications not using unmixing techniques. We have included some discussion of spectral unmixing and the choice of mode-aggregation in the manuscript (L302-307; 359-363).

A comparison of the geomorphology of the reference maps to the corresponding regions in the datasets included in Supplementary Figure 3 would require a more in-depth analysis of the spatial patterns of ~70 mapped regions that differ in alignment and classification scheme, and indeed vary widely across products at similar resolutions. While interesting, this is outside of the scope of this study. We have extended Supplementary Figure 3 to show comparisons of the wetland percentages of each of the regions to both the high-resolution reference maps (~2m) as well as wetland percentages from the coarsened reference maps at the closest resolution to the comparison land cover products. We agree that the implications of Supplementary Figure 3 are interesting, but as mentioned in the manuscript, these findings are based on a very limited sample size (3 sites for the Arctic and 4 for the boreal region) and would require a more robust analysis to infer broader patterns, and so we feel the supplementary is a more appropriate place.

5. There are some places where the language could be easily mis-interpreted – consider revising to be more precise and avoid confusion:

a. Line 118: Consider rephrasing to clarify that this relationship is between estimated/modeled emissions and wetland area since neither variable was “observed” at the regional scale.

Thank you, this has been rephrased as “A strong positive correlation ($R^2=0.6$) was found between regional upscaled estimates of CH_4 emissions and wetland area across the study sites (Fig. 2b).”. (L127-128)

b. Line 126: “change in wetland extent” – consider rephrasing, e.g. “difference in estimated wetland extent” to avoid sounding like a real world change over time.

Thank you. Revised accordingly. (L136)

c. Line 152: “first to exhibit” – consider rephrasing to avoid sounding like a change over time, for example something like “For example, CH_4 estimates from the Kilpisjärvi region transitioned from source to sink at the finest spatial resolution”

Agreed. Rephrased to “...the Kilpisjärvi region exhibited a CH_4 source-to-sink transition at a finer resolution (50 m) relative to the other sites...”. (L163)

d. Line 217: “emissions” should be “emission estimates”

Agreed and revised. (L232)

Minor comments

6. Line 51: Is “potential” necessary here before “emissions”?

Agreed and revised. (L54)

7. Line 52: Should “requires” here be “require”?

Agreed and revised. (L55)

8. Line 55: Please specify uncertainty in what, perhaps “... contribute to the overall uncertainty of CH_4 emission estimates for regions...”

Thank you. Revised to “...contribute to the overall uncertainty of CH_4 emission estimates in a region...”. (L58)

9. Line 56: Remove comma after differences in “for differences, in CH_4 flux ... “

Thank you - Revised. (L60)

10. Line 67-75: Consider specifying, even in parentheses, some of the relevant spatial resolutions or sizes of features mentioned in this section, i.e. how narrow do you mean by “narrow channels” and what are “these resolutions” that “may still be too coarse”.

Thanks for the comment. As the size of features lost depend on the resolution of the imagery, we have rephrased to “Fine-scale landscape features— such as ecotones, narrow channels, patterned ground, and isolated vegetation patches—require spatial resolutions commensurate with their characteristic dimensions (often <10m) in order to be accurately represented, but these are generally obscured or simplified in coarse datasets”. (L75-77)

11. Line 73-75: I’m slightly confused by how the two issues are related in this sentence – I agree both are issues, but wetland-upland boundary mapping seems to be about intra wetland variability from different zones/patches within a wetland site, whereas observational bias towards high-emitting wetlands is at the wetland site scale.

Agreed, there was a missing linkage and caveat here. High-resolution wetland–upland boundary mapping is needed to better quantify candidate areas of CH₄ sinks in or near wetlands, which can in turn reduce underestimation of uptake, historically caused by observational bias— provided uptake is more systematically included in data collection campaigns.

This has been reworded to “areas of CH₄ uptake—underrepresented in current observations— require high-resolution mapping of wetland–upland boundaries to improve estimates of CH₄ sinks in heterogeneous wetlands”. (L81-82)

12. Line 83-86: It may help the reader here to distinguish between spaceborne and airborne sensors, e.g. “... from spaceborne sensors like MODIS... are freely available on a global scale, very high spatial resolution data (<2.5m) available from airborne sensors is often...”.

Agreed and revised. (L90-92)

13. Line 101: Should “variability in spatial resolution” here be just “spatial resolution” or “differences in spatial resolution”? As written this could be interpreted as comparing maps with low variability in spatial resolution to maps with high variability in spatial resolution, and its unclear what that would mean.

Agreed and revised. (L108)

14. Line 141: What does “near” mean in this sentence? Consider rephrasing as “similar to” and/or mention the specific datasets being referred to here.

Agreed and revised. (L148)

15. Line 214: Is there a word missing here? “...should near 25 m”

Thank you – revised. (L229)

16. Figure 1: Treeline boundary line is similar color to the high end of the color palette for wetland distribution. Is the treeline necessary to include? If so, consider using a color more

distinct than the wetland distribution to avoid confusion. The abbreviations (TP, HBL, etc.) also aren't referenced elsewhere in the paper, are they necessary to include?

Thank you. We have changed the color to better distinguish from the wetland coloration and the abbreviations for dominant wetland areas have been removed.

17. Supplementary Table 1: If they are notably different, consider including a column with a brief indication of what methods/type of data each source is based on (e.g. multi-spectral vs hyperspectral, lidar, etc.; any notable differences in classification techniques)

Agreed and revised.

18. How does the highest spatial resolution (2.5 m) resolution compare to the area of chamber measurements?

The chamber areas range from .03 - .36 m². This has been added to the manuscript. (L380)

19. Supplementary Figure 2: What unit is area reported in?

Thank you for the comment. The areas have mixed units (percent × meters) because they result from integrating percent departure (y-axis) across spatial resolution (x-axis). The caption has been updated to “The shaded trapezoidal polygons under each curve indicate the integrated area (%·m) of percent departure across the range of resolutions”.

20. Figure 2b – it is unclear what the error/uncertainty is in this figure. Is this across spatial resolutions?

These are upscaled estimates from only the nominal resolution with the shaded area representing the CI. The caption has been revised to make this clear “Linear regression of upscaled regional FCH₄ (expressed as g C ha⁻¹ h⁻¹) and regional wetland area (%) at the nominal resolution ... Shaded areas represent confidence intervals for the fitted trend.”

21. It would be interesting to see how the pattern shown in Figure 3 compares across sites. Is it possible to include additional lines (different colors?) showing how the sites vary?

Thank you for the comment. Deviation from reference per site is already shown in Supplementary Figure 2.

22. Figure 4: The y-axis scale is somewhat counterintuitive, in that the more sensitive sites have lower sensitivity scores. Consider reversing this axis or perhaps re-naming the y-axis to “rank” instead of “score”.

Thank you – revised.

23. Supplementary Figure 1: This is a very nice figure, however the legends in the bottom row are not readable, at least when printed out. It would be nice to be able to see what classes are included for each site. Is it possible to increase the text size, or separate the legends into a row below the current panels?

Thank you for the comment – the text size and resolution has been increased.

24. Please provide references/more details for the datasets used in Supplemental Figure 3. It would be very useful to also provide the spatial resolution of each, to be able to better link to the study results.

Agreed. The resolutions have been added to the figure and the references have been added to the figure caption and Supplementary References section.

References

Hondula, Kelly L., et al. "Effects of using high resolution satellite-based inundation time series to estimate methane fluxes from forested wetlands." *Geophysical Research Letters* 48.6 (2021): e2021GL092556.

Kuhn, McKenzie, et al. "Emissions from thaw ponds largely offset the carbon sink of northern permafrost wetlands." *Scientific Reports* 8.1 (2018): 9535.

Kyzivat, Ethan D., and Laurence C. Smith. "A closer look at the effects of lake area, aquatic vegetation, and double-counted wetlands on Pan-Arctic Lake methane emissions estimates." *Geophysical Research Letters* 50.24 (2023): e2023GL104825.

Laurion, Isabelle, et al. "Variability in greenhouse gas emissions from permafrost thaw ponds." *Limnology and Oceanography* 55.1 (2010): 115-133.

Prèskienis, Vilmantas, et al. "Seasonal patterns in greenhouse gas emissions from lakes and ponds in a High Arctic polygonal landscape." *Limnology and Oceanography* 66 (2021): S117-S141.

Thornton, Brett F., Martin Wik, and Patrick M. Crill. "Double-counting challenges the accuracy of high-latitude methane inventories." *Geophysical Research Letters* 43.24 (2016): 12-569.

Reviewer #2 (Remarks to the Author):

The manuscript titled "Coarse land cover datasets bias Arctic-Boreal wetland methane budgets" was a unique and novel study. This study provides crucial empirical evidence demonstrating the necessity of high-resolution (≤ 25 m) land cover data with appropriate thematic detail for accurately estimating CH₄ emissions in Arctic-boreal wetland. The findings show that coarse-resolution (> 1 km) maps misrepresent wetland extent and type—particularly high-emitting fens—leading to erroneous source/sink conclusions. This work is highly valuable

for reducing global methane budget uncertainties and sets clear benchmarks for future upscaling efforts. Some specific comments are following:

LN111: Unclear with “at all sites and at four sites”, which four sites?

Agreed. This was originally intended to indicate that four regions (one of which has two sites) had wetlands divided into bogs and fens. This has been revised for improved clarity. “...classes were harmonized to represent total wetlands at all sites. At five sites (Pallas, Tiksi, Seida, and two sites at Scotty Creek), fen and bog classes...”. (L119-120)

LN211: “33.34.”? Please provide correct citation format.

Thank you – Revised. (L225)

LN583: Could you provide another table to show the CH₄ flux of each land cover type of the 7 sites you collected from other studies? It could be a supplementary table to support the result of Figure 2c and Supplementary Figure 1 to see how the regional CH₄ fluxes changed with the alternation of land cover types.

Thank you for the suggestion. This table has been added. (Supplementary Table 2)

LN589/Figure 5: Is the contribution simply based on the area of bog and fen in each spatial resolution class? According to Fig. 2a, the wetland area (e.g. fen and bog) tends to be zero when spatial resolution is larger than 1 km, so CH₄ emissions from wetlands would also be zero. How did you calculate the contribution of fen and bog to FCH₄ at coarser resolution?

Thank you for raising this point. Figure 2a showed median wetland area, where Figure 5 shows mean wetland proportional contribution. The wetland area tended to be zero at the coarsest resolutions (>2.5 km) with the exception of one site that was predominantly wetland at the nominal resolution (Scotty Creek – South), where the contribution of fen and bog come from. The figure has been changed to box plots and language has been added to the manuscript to clarify. (L184-186)

Supplementary Figure 1: The legends in the figure were unreadable. In addition. It would be better to uniform the color of each land cover type across 7 sites. Moreover, lowest spatial resolution here was 1 km, why didn't you show the results of 1250, 2500 and 5000 m so that they were consistent with other figures and tables?

Thank you for the comment. Most of the maps at coarser resolutions are only one class throughout the mapped extent. As the purpose was to conceptually illustrate the change in land cover morphology, we did not include all steps to minimize redundancy. We have harmonized the color palette where possible, increased the legend size, and enhanced the resolution.

Supplementary Table 1: Please uniform the font size below the heading line.

Thank you – revised.

Reviewer #3 (Remarks to the Author):

I thought this was an elegantly put together paper that made a specific point and provided the proper analyses to show what effect the coarseness of remote sensing has on estimated emissions from regional CH₄ fluxes. It makes a compelling case that coarse resolution estimates of the regional flux are almost certainly introducing substantial error. The conclusions seem sound basically everywhere from my interpretation. There are some limits to what the paper can state realistically, but the authors do a good job recognizing those and pointing them out. I also don't think there's an easy way around those limitations. I wouldn't say the results are all that surprising per se, but I'm not aware of anyone who has done this analysis before, and I think it is a useful paper that will be cited frequently. This is a big issue that the field is having to tackle. I have really just a few minor nitpicks that I thought might result in a slightly higher quality presentation.

358-365 - I appreciated the caveats here. One additional caveat that I think should be listed is heterogeneity within any individual wetland (e.g. hydrological variation within the wetland, edge vs center of a particular terrain type, etc). I don't think it affects your analysis at all but it was something I kept thinking about as I viewed your maps.

Agreed. We have added language regarding this limitation. "This approach assumes that each land cover class has a representative flux and does not capture fine-scale hydrological or microtopographic heterogeneity within classes" (L394-395)

Figure 3. Caption should make clearer what shaded area is

Thank you. This has been added to the captions in Figures 2 & 4 and removed from Figure 3 as the standard error is already shown.

Figure 4. You could eliminate y labels and ticks on this graph if the y limits of the panels were lightly adjusted to match those of c

Agreed and revised.

Supplementary Table 1 - It took me a few looks to figure out how to read this table. Could the caption be edited to make clear you need to refer to the supplementary bibliography just to make it more clear? I was looking for footnotes and such at first

Agreed and revised.

Supplementary Figure 1 - The text in the legend is so small it is unreadable in all the files provided for review. I assume there's a vector art image that could be used that would address this. It's a small problem, but should be addressed

Thank you for the comment – the text size and resolution has been increased.